# From Cocoa to Chocolate: Effect of Processing on Flavanols and Methylxanthines and Their Mechanisms of Action

**DOI:** 10.3390/ijms232214365

**Published:** 2022-11-18

**Authors:** Luis Goya, John Edem Kongor, Sonia de Pascual-Teresa

**Affiliations:** Department of Metabolism and Nutrition, Institute of Food Science, Technology and Nutrition (ICTAN-CSIC), Jose Antonio Novais 10, 28040 Madrid, Spain

**Keywords:** cocoa processing, chocolate, flavanol, polyphenol, methylxanthine, molecular mechanism

## Abstract

Despite the health benefits associated with the ingestion of the bioactive compounds in cocoa, the high concentrations of polyphenols and methylxanthines in the raw cocoa beans negatively influence the taste, confer the astringency and bitterness, and affect the stability and digestibility of the cocoa products. It is, therefore, necessary to process cocoa beans to develop the characteristic color, taste, and flavor, and reduce the astringency and bitterness, which are desirable in cocoa products. Processing, however, affects the composition and quantities of the bioactive compounds, resulting in the modification of the health-promoting properties of cocoa beans and chocolate. In this advanced review, we sought to better understand the effect of cocoa’s transformational process into chocolate on polyphenols and methylxanthine and the mechanism of action of the original flavanols and methylxanthines. More data on the cocoa processing effect on cocoa bioactives are still needed for better understanding the effect of each processing step on the final polyphenolic and methylxanthine composition of chocolate and other cocoa products. Regarding the mechanisms of action, theobromine acts through the modulation of the fatty acid metabolism, mitochondrial function, and energy metabolism pathways, while flavanols mainly act though the protein kinases and antioxidant pathways. Both flavanols and theobromine seem to be involved in the nitric oxide and neurotrophin regulation.

## 1. Introduction

Cocoa (*Theobroma cacao* L.) belongs to the family Malvaceae and the genus *Theobroma*. The cocoa tree produces fruits, referred to as pods, along the trunk and branches. The pods are oval and contain the seeds (about 30–40 seeds), commonly known as cocoa beans, which are embedded in a sweet mucilaginous pulp. The pulp is rich in fermentable sugars and high in acidity (pH 3.0–3.5). Chocolate is the most popular cocoa-derived product, highly valued by consumers around the world [1,2,3]. Its consumption has received increased global attention in recent years due to the biologically active components in cocoa beans [1,2,3,4]. However, some adverse effects of chocolate consumption have also been reported, as the final chocolate might be rich in sugar and fat [5]. This limits the use of chocolate as a functional food product and opens a search for more “natural” and unprocessed forms of cocoa. The components of the cocoa bean quality currently used on the international cocoa market include the bean size and mass, moisture content, flavor characteristics, low debris and bean defect levels, fat quality, and content [6,7]. There is a growing global trend in the search for functional attributes in food as quality indicators. In the case of cocoa beans, the final qualitative and quantitative content of the polyphenols and methylxanthines might be considered as part of the quality indicators affecting the final price of the beans.

The present revision will focus on two different types of phytochemicals in cocoa, the main cocoa polyphenols, flavanols, and the foremost cocoa methylxanthine, theobromine, because this compound is as abundant in cocoa as the total of the flavonoids and is currently the subject of an increasing number of studies [8,9,10,11]. Cocoa flavanols and methylxanthines control the molecular pathways that regulate the cell signaling and function in most of the tissues so far investigated. The main pathways studied include the mitogen-activated protein kinase (MAPK) pathways and phosphoinositide-3-kinase-protein kinase B/protein kinase B (PI3K/Akt), which are activated by several ubiquitous growth factors, such as the insulin-like growth factor-1 (IGF-1) and fibroblast growth factor (FGF), among others, and neuro-specific factors, such as the brain-derived neurotrophic factor (BDNF) and nerve growth factor (NGF) [12,13]. Additionally, to the canonical signaling pathways, the systemic factors and molecular/biochemical mechanisms, such as nitric oxide (NO), toll-like receptors, apoptosis, and antioxidant and anti-inflammatory responses, are also specifically involved in the biological effects of cocoa phytochemicals on the tissue and cell function, and these will be discussed in this review. It is worth mentioning that most, if not all, of these pathways are interconnected, and it results in being very difficult most times to unravel and delineate the specific effects. Moreover, several polyphenols included in a diet may affect more than one biomarker or pathway, which may confound conclusions about the specificity of action [12,13,14]. Raw cocoa beans are bitter and astringent and need to be processed to reduce the bitterness and astringency and develop the desired flavors. The bitterness and astringency of raw cocoa beans are due to the phenolics and methylxanthines in the beans [3]. During the cocoa bean processing, a wide range of chemical reactions, including the aldol condensation, polymerization, cyclization, Maillard reaction, and Strecker degradation, which enhance the flavor, color, and shelf stability of cocoa products, have been reported [15,16,17,18,19,20,21]. According to some authors, the polyphenols undergo a biochemical modification through the polymerization and complexation with other compounds such as proteins [6,16], while theobromine and caffeine mainly diffuse into the shells of the beans [22,23]. These result in a decrease in the concentration of the phenolic and methylxanthine compounds and contribute to the reduction in the bitterness and astringency of the beans [22,23]. However, there is no general agreement regarding the negative effect of food processing on the content of the bioactive compounds in processed cocoa products. In fact, as occurs with other processed foods, some authors have shown that processing, far from reducing the biological activity of foods of plant origin, could enhance their bioavailability and, beyond that, their bioactivity [24,25]. Moreover, during processing, notably the fermentation and drying processes, flavor precursors (free amino acids and reducing sugars) are formed, and these combine during the roasting process through a complex series of thermal reactions, mainly the Maillard reaction and Strecker degradation, and are converted into desirable aroma volatiles, such as pyrazines, aldehydes, ketones, furans, and pyrroles [18,19,20,21]. Nevertheless, cocoa processing clearly affects the flavanols and methylxanthine content of the final product and therefore will affect its potential biological activity and the molecular mechanisms that mediate it.

The purpose of this review is to provide a better understanding of the various processing steps of cocoa beans, from harvesting to chocolate production, emphasizing the changes in the main bioactive compounds in cocoa: the polyphenols (mainly flavanols) and methylxanthines (mainly theobromine) during processing that will affect the mechanism of action implicated in their biological activity. The research gaps are also identified to aid in the planning of future studies. Because this review aims to be an update of the information regarding the flavanols and methylxanthines of cocoa, most of the results reported up to 2020 will be briefly summarized and largely referred to comprehensive reviews; henceforth, we will mainly focus on recent data from the last couple of years, in which, despite the pandemic, research on this topic has been rather productive.

## 2. Cocoa Bean Processing

The processing of cocoa beans can be categorized into primary and industrial processing. The primary processing of cocoa entails all the processes cocoa pods go through, from harvesting to obtaining the final dried beans (Figure 1). They play a significant role in the development of the final color, flavor profile, and composition of the bioactive compounds of the dried beans as well as the shelf stability of the beans during transportation and storage [26,27]. These include harvesting, pod storage (as a means of pulp pre-conditioning), pod breaking, fermentation, and drying. Although cocoa bean quality is greatly influenced by the genetic makeup and origin of the cocoa, inadequate or poor primary processing could result in beans with a poor quality and shelf stability.

The industrial processing of cocoa can also be sub-classified into secondary and tertiary processing. The secondary processing involves all the processes the dried fermented beans are taken through to obtain semi-finished products, such as cocoa liquor (cocoa mass), cocoa butter, cocoa powder, etc. The secondary processing includes cleaning, breaking and winnowing, sterilization, alkalization, roasting, nib grinding, and liquor processing (Figure 1). The tertiary processing involves the use of the semi-finished cocoa products and other ingredients to produce chocolates, cocoa/chocolate beverages, and other confectionery products.

### 2.1. Primary Processing of Cocoa Beans

#### 2.1.1. Harvesting

The harvesting of ripe, matured cocoa pods initiates the primary processing of cocoa. For optimal processing and the production of high-quality beans for cocoa-based products, only matured, ripe, or at least semi-ripe, and disease-free pods are harvested. The Studies have reported on the direct influence of harvesting cocoa pods at an appropriate maturity stage on the flavor characteristics of the cocoa beans [28]. The cocoa beans obtained from unripe pods have been found to have low sugar contents and do not ferment properly, while over-ripe ones, on the other hand, are easy to germinate and be infected by microorganisms [29,30]. In terms of the bioactive compounds in cocoa beans, studies have shown that cocoa beans obtained from immature or over-ripe fruits contain fewer bioactive compounds than beans from fully mature fruits [30,31]. A recent study by Dang and Nguyen [32] showed that the maturity at harvesting and the fermentation conditions significantly increased the flavanol and methylxanthine contents, together with the antioxidant capacity of the cocoa beans (Table 1). Earlier studies by Zheng et al. [33] and Pereira-Caro et al. [30] found the theobromine and caffeine contents appeared at maturity when the cocoa beans started to develop their seed coats, which continued to increase until the cocoa fruits were fully ripe. Zheng et al. [33] also explained that the increase in the alkaloid compounds could be due to their transport from the pericarp to the cocoa bean during fruit maturation, resulting in their increased concentration in the beans of matured fruits. Pereira-Caro et al. [30] noted that because the alkaloid compounds were located in the same storage cell as the phenolics, they could develop together with the phenolics during maturation.

#### 2.1.2. Pod Storage

The cocoa pulp is the substrate sequentially metabolized by microorganisms during the fermentation process. The properties of the substrate determine the type and quantities of the microbial development and metabolism during the fermentation, hence changes in the pulp may affect the production of alcohols by the yeasts and the subsequent production of acids by the lactic acid and acetic acid bacteria [6]. Pod storage (PS) is storing harvested cocoa pods for a period before opening the pods and fermenting the beans. The harvested cocoa pods are living tissues and undergo metabolic activity, such as respiration and transpiration, using the sugars in the pulp. This activity leads to decreased sugar content in the pulp for yeasts to utilize during the start of the microbial phase of fermentation, resulting in reduced alcohol production by the yeasts. The reduced alcohol production in turn results in a reduction in the acetic acid production by the acetic acid bacteria, which will diffuse into the beans to initiate the biochemical reactions. The post-harvest storage of cocoa pods has been reported to reduce the nib acidification during the subsequent fermentation, result in a reduction in acid notes, and increase the cocoa flavors in the resulting cocoa beans [26]. The pod storage of cocoa has also been found to cause a significant reduction in the polyphenolic and methylxanthine content of cocoa beans after the fermentation and drying (Table 1), thereby reducing the astringency and bitterness in the cocoa and cocoa products.

#### 2.1.3. Pod Breaking

In general, the pod breaking involves opening the harvested pods to extract the wet beans for fermentation, and it is important not to damage the beans in this process. Cutting the beans will allow insects or molds to enter the bean.

#### 2.1.4. Cocoa Bean Fermentation

Cocoa bean fermentation is crucial in making cocoa beans more valuable and stable. It also influences the marketability and acceptability of the beans. It is conducted for the development of the characteristic brown cocoa color and modification of the polyphenolic and pH, leading to a reduction in the astringency and bitterness of the final dried beans. The fermentation also generates the flavor precursors, namely the free amino acids and peptides, from the enzymatic degradation of the cocoa proteins, reducing sugars from the enzymatic degradation of the sucrose [27] from which the typical cocoa aroma is generated during the subsequent roasting process.

During the fermentation, the beans are removed from the pods and subjected to various activities of various microorganisms, which are prevalent within the surrounding environment. The changes in the cocoa beans produced by the fermentation occur in two stages: the microbial fermentation, which takes place in the pulp, and biochemical changes, which occur in the beans following the death of the embryo. Following the death of the embryo and the breakdown of the cell walls in the bean, endogenous seed enzymes (e.g., proteases, polyphenol oxidases, glycosidase, and invertase) and substrates (e.g., polyphenols, proteins, and sugars) interact and react in a specific manner [18,61].

The bioactive compounds in the beans are subjected to a biochemical modification through the polymerization and complexation with protein, hence decreasing the solubility and astringency [6]. Polyphenols generally decrease as the fermentation degree increases (Table 1). The activity of the polyphenol oxidases (PPOs) during the fermentation is associated with the reduction in the polyphenols [18]. Studies have also shown that the loss of polyphenols during fermentation is not only due to the oxidation process but also caused by the diffusion of the polyphenols into fermentation sweating [62]. The methylxanthine contents of cocoa beans have also been found to diffuse into the shells of the beans during fermentation, thus causing a reduction in the concentration and contributing to the reduction in the bitterness of the beans [22,23]. The purple-colored anthocyanins located in the specialized vacuoles within the cotyledon are hydrolyzed by glycosidases to anthocyanidins [63]. Thompson et al. [64] observed that the enzyme cleaves the sugar moieties galactose and arabinose attached to the anthocyanins. This results in the bleaching of the purple color of the beans to brown as well as the release of reducing sugars that can participate as precursors to reactions during the roasting. In general, different cocoa types require different degrees of fermentation, but usually, it takes from 4 to 7 days.

#### 2.1.5. Drying of Fermented Cocoa Beans

The moisture content of the cocoa beans after fermentation is about 60%, and this needs to be reduced to around 7% by drying [65,66] to prevent a mold infestation and the subsequent formation of Ochratoxins during storage. Cocoa bean drying is, however, more than just reducing the moisture content of the beans as it also allows some of the chemical changes, which occurred during the fermentation stage, to continue and improve the bean color and flavor development [42,67]. During the drying of fermented cocoa beans, the active polyphenolic oxidase catalyzes the polyphenols oxidation reactions into quinones, which are subjected to further condensation with free amine and sulfhydryl groups, which leads to the synthesis of brown polymers, development of new flavor components, and loss of membrane integrity [6]. This helps to further reduce the bitterness and astringency and develop the chocolate brown color of well-fermented cocoa beans. The timing of the drying process is crucial as the rapid drying of the beans results in case hardening which prevents the outward migration of acetic acid from the beans, thus leading to a build-up of acidity in the beans [68,69]. On the other hand, if the drying is too slow, molds and off flavors can develop [68,70].

### 2.2. Industrial Processing of Cocoa Beans

#### 2.2.1. Cleaning

Cocoa beans for processing are first evaluated for quality, which is important to the final product quality. The beans are then subjected to a cleaning process which involves the removal of all the extraneous materials from the cocoa beans. The process involves a sieving operation to remove all the extraneous materials, such as stones, strings, coins, wood pieces, soil particles, nails, etc. Cleaning before processing is essential to ensure the production of quality products devoid of extraneous materials and to also reduce the wear on machinery, as pieces of metal or stone can cause extensive abrasion on some of the equipment [15,71].

#### 2.2.2. Breaking and Winnowing

Processing cocoa beans into semi-finished products, such as cocoa liquor, powder, and butter, requires the use of the essential part of the beans: the nibs (cotyledon). Thus, the nibs need to be separated from the shells. Breaking and winnowing entail applying pressure on the cocoa beans to loosen the shells (breaking) and subsequently separating the shells from the nibs so that only the nibs remain (winnowing). A thorough separation of the shells from the nibs is necessary to ensure a high-quality product devoid of a bitter and unpleasant taste with a gritty texture. Breaking and winnowing can be performed after roasting the beans (the traditional process) or before roasting. When the latter is performed, the nibs can easily be treated with an alkali solution (alkalization) [72].

#### 2.2.3. Sterilization

Cocoa beans are exposed to microbial contamination during the primary processes of fermentation, drying, bagging, and transportation. Cocoa beans or nibs are thus subjected to saturated steam under pressure for a sufficiently long time to reduce the microbial load [15]. This process is referred to as sterilization and can be performed before or after roasting. In the latter situation, sterilization after roasting is used to ensure the destruction of heat-resistant bacteria and spores that might have survived the high temperatures of the roasting process [15,73]. During sterilization, an exposure time of seconds, rather than minutes, in the hot, low-pressure steam is sufficient to sterilize the bean [73]. This might be insufficient to cause any change in the flavanols and methylxanthine content. No work has reported changes in the flavanols and methylxanthine contents during sterilization.

#### 2.2.4. Alkalization (Dutching)

Alkalization, also known as Dutching, is an optional step in the processing chain where cocoa nibs are treated with an alkali solution, such as potassium bicarbonate or sodium bicarbonate, to increase the pH to 7–8 with the purpose of modifying the color and taste and improve the dispersibility of the powder solids in water [15,20,45,54,74]. Alkalization has been found to influence the composition of the flavonoids and methylxanthines in cocoa products [45,75]. Alkalization has been found to result in the oxidation and polymerization of flavonoids, leading to a reduction in the astringency in alkalized cocoa nibs and powder [20,47]. High losses of polyphenols and changes in their composition have been reported in alkalized cocoa nibs and powder [20,42]. Work by Urbańska et al. [42] observed over a 60% loss of total polyphenols, and according to Giacometti et al. [47], the greatest losses were observed for epicatechin and catechin (up to ca. 98 and 80%, respectively) as well as for quercetin (ca. 80%). Earlier studies by Andres-Lacueva et al. [24] reported a reduction of 67% of epicatechin and 38% of catechin in cocoa powder after alkalization. Alkalization has also been found to decrease the amount of methylxanthine in the cocoa nib, thereby reducing the bitterness of alkalized cocoa powder [20,21,53]. The reduction in the methylxanthine content increases with an increasing degree of alkalization, and Li et al. [38] observed the greatest losses for theobromine (over 20%).

Alkalization can be performed in three main ways which include: nib alkalization, cake alkalization, and liquor alkalization [72,76]. Nib alkalization is the most common alkalization method, and it involves subjecting the cocoa nibs to an alkali treatment before the roasting and subsequent milling into liquor [72]. In cake alkalization, the cocoa nibs are first roasted and milled into liquor. The liquor is then pressed to extract the cocoa butter and the resultant cake is then kibbled (i.e., broken down into gravel-sized pieces) and subjected to an alkali treatment in a reaction vessel. Liquor alkalization involves subjecting the cocoa liquor to an alkali treatment to modify the color and flavor.

#### 2.2.5. Cocoa Bean Roasting

Cocoa bean roasting involves subjecting the dried fermented cocoa beans to high temperatures, usually between 120 and 150 °C [47,77,78] for 15–45 min [79], and it is an important technological step that results in the production of flavor and aroma compounds, as well as the color changes that are desirable in cocoa products [6,42,77,80]. The time–temperature combination used in roasting depends on factors such as the type of cocoa (Criollo or Forastero), the intended final product, and the cocoa material (whole beans, nibs, or liquor) [69]. Several physiochemical changes occur during cocoa bean roasting due to the heat penetration into the beans. These changes include the loss of moisture from the beans (to ~2%), evaporation of volatile acids that contribute to acidity and bitterness, and loosening of the bean structure, which facilitates the removal of the shells and pressing of the butter during the liquor processing [15,42]. The heat also serves as a sterilization treatment which results in a further reduction in the number of microorganisms present in the beans [15,80].

Roasting significantly affects both the amount and composition of the polyphenols in cocoa beans [42,49,81]. It reduces the amounts of the total polyphenols, (−)-epicatechin, and proanthocyanidin in cocoa beans [50,51]. Żyżelewicz et al. [49], however, found increased catechin content during 20 min of roasting at 135 and 150 °C, which was attributed to the epimerization of the flavan-3-ol monomers and proanthocyanidins. Polyphenols are known to be thermolabile molecular compounds, and their content decreases with high temperatures and a prolonged roasting time. Ioannone et al. [82] believed that a low temperature and short roasting duration are better to preserve the polyphenolic content, with a concomitant improvement in the antioxidant activity of the roasted beans [51]. Temperatures below 140 °C have thus been recommended by Urbańska et al. [42] to preserve the polyphenolic content. Roasting has also been found to reduce the alkaloid content in cocoa beans. Aprotosoaie et al. [48] explained that during roasting, theobromine and caffeine could bond with diketopiperazines, resulting in a lower concentration of free alkaloids.

Cocoa beans can be roasted in three different ways, namely whole beans, nib, or liquor/mass roasting [15]. Cocoa beans are usually roasted using the whole-beans roasting method to produce the cocoa mass/liquor. This facilitates the removal of the shells during the subsequent breaking and winnowing stages. Nib roasting involves the removal of the shells before subjecting the nibs to roasting. Removing the shells before roasting enables the nibs to be treated with an alkali solution to change the color of the final cocoa powder or be treated with water or a sugar solution for flavor enhancement [15,20]. Liquor/mass roasting involves removing the shells and grounding them into liquor. The liquor/mass is then roasted, and the roasting can be performed using relatively inexpensive and simple equipment employing a scraped surface heat exchanger.

#### 2.2.6. Cocoa Nib Grinding and Liquor Processing

Cocoa nibs are milled into a low-viscous mass referred to as cocoa liquor or cocoa mass. The viscosity of the liquor produced is greatly influenced by the degree of roasting and the moisture content of the nib. Cocoa liquor consists of the cocoa solid particles suspended in the cocoa butter [83]. The nibs are ground in a two-stage process: coarse grinding and fine grinding. The coarse or pre-grinding involves transforming the nibs from a solid to a thick paste, and this can be carried out in hammer mills, disc mills, or pin mills. Fine grinding, on the other hand, is performed to transform the thick paste into a smooth low-viscous mass, and it determines the end fineness and quality of the cocoa liquor. It is carried out in bead or ball mills. In small-scale processing, a kitchen blender can be used for coarse grinding, while the melanger is used for fine grinding. The desired end fineness of cocoa liquor is reported to be 15–70 μm [84] or 99.5% of the particles should be ≤75 μm [85]. Grinding cocoa nibs into liquor involves the application of shear under a controlled temperature. Changes in the polyphenols and methylxanthine contents during grinding have not been reported. Future studies can focus on the effects of the coarse and fine grinding of cocoa nibs on the polyphenols and methylxanthine composition of the final liquor produced. Cocoa liquor can be used directly in the manufacture of chocolate or processed by pressing to obtain two main products—cocoa butter and cocoa pressed cake (with a fat content between 10 and 24%) [15]. The extracted cocoa butter is used in the manufacture of chocolate and pharmaceutical and cosmetic products, whereas the cocoa press cake is further broken down into kibbles and pulverized to form cocoa powder. The pulverized cocoa powder is sieved to obtain a powder with a standardized particle size which is then used as the main ingredient in cocoa drinks and/or chocolate-flavored beverages. Anoraga et al. [86] noted that the yield of cocoa liquor pressing was influenced by the temperature, cocoa beans moisture content, particle size, and pressing time. However, the extent to which the bioactive compounds in cocoa liquor are affected during pressing remains unclear as no study has reported on it yet.

### 2.3. Chocolate Production

Chocolates are semisolid suspensions of fine solid particles from sugar and cocoa (and milk solids, depending on the type) in a continuous fat phase [15]. The conventional or industrial chocolate manufacture generally consists of the mixing of ingredients (the cocoa liquor/masse, sugar, cocoa butter, and milk component, depending on the chocolate type), refining (to reduce the particle size of the mixed chocolate paste), conching of chocolate paste, tempering (to form the stable fat crystals in the cocoa butter), casting and molding, cooling and de-molding, and finally, wrapping and packaging. The quality characteristics, such as the melting and rheological behavior, flavor, and sensory perception of chocolates as well as the bioactive compounds compositions, are influenced largely by the ingredient composition and processing method used [87].

One of the chocolate processing steps which could have a probable influence on the composition of the bioactive compound is conching. Conching consists of the mixing, shearing, and aeration of the chocolate mass during heating at a certain temperature (usually > 40 °C) to produce liquid chocolate, where all the solid particles are coated with fat [48,88]. It is an important stage for improving the quality of the chocolate. The main purpose of the conching process is to thoroughly combine all the ingredients to obtain a homogeneous mass [89]. It is also performed to evaporate the residual volatile compounds (e.g., acetic acids) and moisture to improve the color, texture, and flow characteristics of the chocolate mass [90]. Different time/temperature combinations are used in the conching process, depending on the chocolate type. Konar et al. [91] and Owusu et al. [92] have noted that temperatures ranging from 70 to 90 °C can be used to conch dark chocolates, while other studies have reported conching temperatures between 50 and 60 °C for milk chocolates to avoid the formation of Maillard compounds [89,93,94].

Limited studies have been conducted on the influence of conching on the polyphenolic and methylxanthine content of chocolates and their antioxidant properties, with varying opinions. Barišić et al. [52] noted that volatile polyphenols are lost during the initial stage of conching due to evaporation, together with water and short-chain fatty acids. Earlier studies by Afoakwa et al. [56] established that the content of the volatile polyphenols is reduced by 80% in this process. Sulistyowati and Misnawi [55] also observed a significant decrease in the concentration of the polyphenol and antioxidant activity due to the conching temperature. However, other studies found no significant variations (3%) in the phenolic content and pattern, as well as the antioxidant activity during the conching, regardless of the time/temperature combination applied [58,59,60]. The same results were reported by Di Mattia et al. [57] for the total polyphenol content.

Tempering is one of the most critical processing steps for making quality chocolate. It involves cooling while stirring the chocolate mass derived from the conching (from 40–50 to 18–28 °C) to obtain a stable form of the crystalline fat (polymorphic form V) responsible for the good melting properties and the glossy surface of good-quality chocolate [42,95]. However, changes in the polyphenols and methylxanthine contents at this stage have not been reported. Studies are needed to understand the influence of tempering on the phenolic and methylxanthine composition of chocolates.

## 3. Mechanisms of Action of Cocoa Flavanols

### 3.1. Recent Advances in Cocoa Flavanols on Signaling Pathways

In the last two decades, an emerging topic has been the study of the effect of flavonoids on MAPK and upstream/downstream-related proteins, receptors, or enzymes that control the cellular processes, such as proliferation, differentiation, apoptosis, and stress responses, under both normal and pathological conditions. MAPK signaling is activated in response to intra- and extracellular signals that activate the transmembrane glycoproteins of the tyrosine kinase receptor type, leading to the regulation of target genes. Three MAPK families have been reported in mammalian cells: extracellular signal-regulated kinase (ERK), c-Jun N terminal kinase/stress-activated protein kinase (JNK/SAPK), and p38 kinase [6,7,8]. The cyclic AMP-responsive element-binding protein 1 (CREB), a ubiquitous transcription factor, is activated by phosphorylation and is involved in the regulation of many cell processes [12,13,14]. The family of PI3Ks are enzymes involved in cell growth, proliferation, differentiation, survival, and intracellular trafficking. The activation of PI3K generates phosphatidylinositol (3,4,5)-trisphosphate (PIP3) and phosphatidylinositol (3,4)-diphosphate. Upon activation by upstream effectors such as PIP3 or phosphoinositide-dependent kinase-1 (PDK1), Akt translocates to the cell membrane and induces an enhancement of the mammalian target of rapamycin (mTOR) and the stimulation of cellular activities. Another family of signaling proteins involved in metabolic regulation that are regulated by flavonoids is the sirtuins (SIRT), especially SIRT1, that function as histone deacetylases [12,13,14].

The specific effect of cocoa flavanols, catechins, and procyanidins on MAPKs and PI3K/Akt signaling pathways has been frequently reported in different cell types and tissues [96]. Pioneer studies showed that epicatechin plays a role in liver cell survival, partially mediated by the induction of the AKT/PI-3-kinase and ERK1/2 pathways at micromolar concentrations [97,98]. A similar result was observed when a phenolic extract of cocoa was tested; thus, cocoa flavanols upregulated the antioxidant enzyme activity via the ERK1/2 pathway to protect against oxidative stress-induced apoptosis in hepatoma HepG2 cells [99]. Simultaneously, the same authors demonstrated that epicatechin-induced NF-κB, activator protein-1 (AP-1), and nuclear transcription factor erythroid 2p45-related (Nrf2) via the PI3K/AKT and ERK signaling in the same cell line [100]. A year later, the same group showed that cocoa procyanidin B2 and a cocoa polyphenolic extract inhibited the acrylamide-induced apoptosis in human colonic Caco-2 cells by the activation of the JNK pathway [101]. In the same human colonic cells, procyanidin B2 induced the Nrf2 translocation and glutathione-S-transferase P1 expression via ERKs and p38-MAPK pathways [102]. A year later, it was reported in cultured liver hepatoma cells that cocoa flavonoids improved the insulin signaling and repressed the glucose production [103] and protected those cells against high glucose-induced oxidative stress [104] via AKT and AMPK.

Because pure phenolic compounds are unlikely to avoid metabolism before reaching the cells, the study of cocoa flavanols metabolites has been an emerging topic during the last decade, in particular colonic microbiota metabolites from catechin and epicatechin. Thus, it has been later reported that microbial phenolic metabolites from cocoa flavanols improved the glucose-stimulated insulin secretion via ERKs and PKC pathways [105]. More recently, it has been reported that epicatechin and its colonic metabolite 3,4-dihydroxyphenylacetic acid protects the renal proximal tubular cell against high glucose-induced oxidative stress by modulating the NADPH oxidase (NOX)-4/SIRT1 signaling [106]. Most of that information regarding the ERK 1/2/CREB/TrkB-PI3K/Akt/mTOR signaling on all polyphenols, including cocoa flavanols, has been recently updated in comprehensive reviews [13,14,107]. New data have since been added to this particular topic that seem to mainly support previous evidence on the unambiguous effect of flavanols on these crucial signaling pathways. Thus, *Camellia fascicularis* extract, one of whose major components is epicatechin, could markedly inhibit the phosphorylation of p65, ERK, and JNK, thereby suppressing the activation of the NF-κB and MAPK signaling pathways, which may induce the secretion of pro-inflammatory cytokines [108].

Recent data also claim flavanols, through MAPK and related pathways, are potent inhibitors of neuroinflammation [109]. A recent study has reported that anti-inflammation properties of epicatechin on cultured macrophages were achieved, at least partly, by inhibiting the phosphorylation of the signal proteins (p65 and p38) involved in the MAPKs/NF-κB pathways [110]. In another recent study of cocoa flavanols, epicatechin and catechin have shown a significant docking capacity to TLR-4, JNK, NF-kB, and AP-1 through the formation of multiple hydrophilic and hydrophobic interactions [111]. Finally, epicatechin plays a protective effect on cardiac fibrosis, preventing myofibroblasts transformation, a process that involves the activation of the sumoylation of SIRT1 through SP1. Furthermore, SIRT1 inhibited the Ang II-induced fibrogenic effect via the AKT/glycogen synthase kinase (GSK3b) pathway [112]. Despite this plethora of data in support of the specific effect of flavanols on major signaling pathways and its potential preventive/therapeutic benefit in life-threatening diseases such as cancer or diabetes, the reported effect of cocoa flavanols has been inconceivably obviated in recent reviews on health effects of diet polyphenols [113,114,115,116,117,118].

### 3.2. Recent Advances in Cocoa Flavanols on Nrf2 Pathway/Antioxidant Defenses and Inflammatory Process

Although polyphenols exert their antioxidant capacity mainly through the direct neutralization of free radicals and chelating metals, such as Fe2+ and Cu+ [119,120], there are other mechanisms that have been reported, such as the stimulation of mitochondrial biogenesis through the activation of the SIRT1 and Nrf2 signaling pathways [120,121]. Nrf2, a transcription factor that regulates antioxidant responses, remains in the cytoplasm bound to the Kelch-like ECH-associated protein 1 (Keap1). Oxidative stress and some bioactive products disrupt the binding to Keap1 and release Nrf2 to translocate into the nucleus, where it binds to the antioxidant response element (ARE) in the promoter region of many antioxidant genes and initiates their transcription [121]. These antioxidant proteins include phase I antioxidant defense enzymes; phase II drug-metabolizing enzymes, such as glutathione-S-transferase (GST), NAD(P)H-quinone oxidoreductase-1 (NQO1), hemeoxygenase-1 (HO-1), and UDP-glucuronosyl transferase (UGT); or phase III transporters (multidrug resistance-associated proteins (MRPs) [121].

In particular, cocoa and its flavonoid compounds exert their protective effect against oxidative stress by targeting the transcription factor Nrf2 and Keap1, which participate in the regulation of the ARE. Thus, the regulation of Keap1 can lead to the nuclear accumulation of Nrf2 and the subsequent ARE activation [96]. The first reports on the role of cocoa flavanols on Nrf2 described that epicatechin increased the reduced glutathione (GSH) content, stimulated Nrf2 via the AKT/PKB in the astrocytes [122], and induced the Nrf2 translocation and phosphorylation in cultured hepatoma cells [100]. Similarly, procyanidin B2 evoked a dose-dependent increase in the glutathione peroxidase (GPx), glutathione reductase (GR), and GST, which improved the antioxidant response to an oxidative challenge in colonic Caco-2 cells [102]. In addition, procyanidin B2 induced the Nrf2 translocation and GST P1 expression to protect human colonic Caco-2 cells against oxidative stress [102]. Finally, catechin decreased the lipid peroxidation and reactive oxygen species (ROS) and increased the activity of GPx, the GR total sulfhydryl groups, and the expression of Nrf2 and heme oxygenase-1 in intestinal Int-407 cells [123].

Later, results on mouse cortical neuron cultures suggested that a combination of epicatechin and quercetin activated the Akt- and Ca(2+)-mediated signaling pathways that converge on nitric oxide synthase (NOS) and CREB; this effect results in synergistic improvements in the neuronal mitochondrial performance which confer the profound protection against ischemic injury [124]. Ramírez-Sánchez and colleagues [125] have demonstrated that epicatechin reverses the negative effects that high glucose or simulated type 2 diabetes has on NOS function in the diabetic heart. More recently, cocoa catechins were shown to improve the cellular redox state, resulting in the Nrf2 nuclear migration and upregulation of genes critical for mitochondrial respiration, glucose-stimulated insulin secretion, and ultimately improved the β-cell function [126]. Most of these previous results have been recently reviewed [127,128], although data concerning cocoa flavanols have been ignored in other reviews [129]. A recent study has reported that epicatechin reduced the cardiac fibrosis in an aged female rat model of pre-heart failure, which correlates with significant reductions in oxidative stress and cytokine levels in the absence of changes in contractile function [130]. Another study by Daussin et al. [131] also showed that the administration of cocoa flavanols to mice for 15 days stimulated the NAD metabolism which enhanced the SIRT metabolism and improved the mitochondrial function [131]. In addition, supplementation with a cocoa–carob blend diet rich in flavanols to Zucker diabetic fatty rats counteracted the oxidative stress in diabetic hearts by downregulating the NADPH oxidases, reducing the ROS generation, and modulating the SIRT1/Nrf2 signaling pathway, overall improving the antioxidant defense. Moreover, the supplemented diet suppressed the inflammatory and fibrotic reactions by inhibiting the NF-kB and pro-inflammatory and pro-fibrotic cytokines [132]. While testing a flavanol-rich cocoa supplementation during training on exercise performance, the results showed that the oxidative stress was lower in the cocoa-treated group than in the control group, together with lower interleukin-6 levels, an effect that might be mediated by the decrease in the expression of nuclear factor Nrf2 [133]. Finally, the administration of a 10% cocoa-enriched diet for 25 days to rats was able to prevent the excessive oxidative stress induced by intensive exercise, although it was not enough to avoid the immune function impairment due to exercise [120].

### 3.3. Recent Advances in Cocoa Flavanols on Cognitive Function

Cognitive function is defined as the mental performance that enables information processing, applying knowledge, and changing preferences [134]. Cognitive capacities, especially memory, attention, execution, and processing speed, gradually deteriorate throughout the adult lifespan, and lifestyle tactics such as diet may represent a favorable opportunity to delay or prevent the progressive cognitive decline [135,136]. In this context, numerous pieces of evidence from human clinical studies have strongly suggested that cocoa and cocoa-derived product consumption can be an effective, safe, and attractive approach to improving general cognition and working memory, especially among older people at risk or with cognitive decline [137,138].

Perhaps the first approach in the field was the finding of the uptake and metabolism of epicatechin and its access to the brain after oral ingestion [139]; then, the effect of flavanol-rich cocoa on the functional magnetic resonance imaging (fMRI) response to a cognitive task in healthy young people was reported [140]. Upon the discovery of the translational control by MAPK signaling in long-term synaptic plasticity and memory [141], Schroeter and colleagues [142] observed that epicatechin stimulated an ERK-dependent cyclic AMP-response element activity in the cortical neurons; this positive effect of epicatechin on MAPK was later confirmed in other tissues [104]. Perhaps due to this molecular mechanism, intervention assays in humans, such as the CoCoa Study, showed significant benefits in the cognitive function, blood pressure, and insulin resistance through cocoa flavanol consumption in elderly subjects with mild cognitive impairment [143]. A year later, in a randomized controlled trial (RCT), a very low dose of cocoa polyphenols enhanced positive mood states but not cognitive performance [144], and a positive effect of flavanol-rich cocoa on cognitive capacity was further observed on cerebral perfusion in healthy older adults during the conscious resting state [145]. Considering all previous data, it is now widely assumed that cocoa monomeric flavanols, catechin, epicatechin, and their microbial metabolites, cross the blood–brain barrier and localize in the brain areas connected to learning and memory, such as the hippocampus, cerebral cortex, cerebellum, and striatum, which could potentially lead to cognitive enhancement. All these reports were exhaustively reviewed in 2020 [146,147,148,149,150,151] and more recently last year, with a focus on the molecular mechanisms and cognitive endpoints [13].

Ever since these comprehensive reviews, some new studies have mostly confirmed or substantiated the beneficial effect of cocoa flavanols on cognitive function. Epicatechin supplementation prevented short-term recognition memory impairment in high-fat diet-induced obese mice [152]. In a human study with healthy male volunteers, acute flavanol intake improved the efficiency of blood oxygenation (amplitude and speed) during hypercapnia in the frontal cortical areas of young healthy subjects; this is likely to contribute to improvements in cognitive function, but only when cognitive demands are high [153]. Additionally, in a crossover RCT, where 30 healthy men ingested a cocoa flavanol beverage (high-flavanol 150 mg vs. low-flavanol < 4 mg epicatechin) 1.5 h before an 8 min mental stress task, the flavanols were effective at counteracting the mental stress-induced endothelial dysfunction and improving the peripheral blood flow during stress [154]. In a recent comparative study between (−) and (+) epicatechin, on the mouse frontal cortex, both enantiomers, but more effectively (+)-epicatechin, upregulated the neurogenesis markers, likely through the stimulation of the capillary formation and NO triggering, resulting in improvements in memory [155]. Earlier this year, a randomized, double-blind, parallel-group study was performed on 60 healthy volunteers between 50 and 75 years old who consumed cocoa powder, and the results showed an improvement in executive function, without any change in neurotrophin levels [156]. In a recent review on cocoa flavanols and the aging brain, the authors concluded that neuropsychological ameliorations after cocoa intake were preceded by increases in the cerebral blood flow [157]. These results are in line with those of animal experimentation because improvements have been found in the motor and spatial performances of young and aging mice or rats as well as animal models of Alzheimer’s disease and Parkinson’s disease [158]. Considering that normal age-related memory decline is now considered an impending cognitive epidemic, dietary cocoa flavanols may offer meaningful benefits to cognitive health.

Nonetheless, cocoa/flavanol/chocolate treatment has also proved ineffective for cognitive function in some recent studies. In an eight-week RCT (FlaSeCo study), the short-term use of dark chocolate naturally high in flavanols showed no benefit in the studied cognitive parameters in cognitively healthy older adults [159]. In addition, the most recent clinical assay, a COSMOS-mind study with over 5000 participants, has concluded that a daily intake of cocoa extract for 3 years had no effect on cognition [160]. Moreover, over a median follow-up of 11.8 years, the results of the Framingham offspring cohort failed to declare a clear association between flavanol intake and a slower decline in cognitive function [161]. Despite these disappointments, most recent reviews have strongly claimed an overall enhancement of cognitive function by cocoa flavanols. In a systematic review/meta-analysis published in 2021, Gardener and colleagues [162] reported that of the 15 studies reviewed, 11 (2 observational studies, 6 chronic, and 3 acute intervention studies) found improvements in at least one cognitive domain following flavanol consumption. Of these 11 studies, flavanol intake was associated with improvements in global cognition as well as the cognitive domains of visual-spatial memory and organization, working memory, abstract reasoning, accuracy, reaction time, executive function, episodic memory, verbal fluency, and recognition memory [163]. A comparable conclusion is reached in recent systematic reviews and meta-analyses on polyphenols and cognition in humans [163]. It seems that cocoa’s long-term cognitive protection could particularly affect populations at risk or with early cognitive decline compared to old people who are cognitively intact [164,165]. However, we should bear in mind the fact that sugar content in chocolate and cocoa products is in general not declared and could in fact be a confounding factor in neurocognition studies. Sugar could affect cognition function and thus the inclusion of sugar-controlled studies would be desirable.

### 3.4. Recent Advances in Cocoa Flavanols in Cardiovascular Function

Perhaps the best established benefit of cocoa flavanols on health is their positive effect on cardiovascular function; in fact, the European Food Safety Authority (EFSA) has published two claims in support of the bioactivity of cocoa flavanols: cocoa flavanols help maintain normal blood pressure [166] and endothelium-dependent vasodilation [167], which contribute to normal blood flow. Moreover, to obtain the beneficial effect, a daily intake of 200 mg of cocoa flavanols is recommended, a quantity provided by 2.5 g of high-flavanol cocoa powder or 10 g of high-flavanol dark chocolate, both of which can be consumed in the context of a balanced diet. A recent meta-analysis provides evidence that cocoa flavanols could significantly improve endothelial function, with an optimal effect observed with 710 mg total flavanols, 95 mg (−)-epicatechin, or 25 mg (+)-catechin [168].

An overall agreement on the beneficial effects of cocoa flavanols on cardiovascular function has been proclaimed since the pioneer studies from two decades ago [169,170,171,172,173,174,175], through the first human trials [176,177,178] and first meta-analysis of controlled trials [179,180,181], up to the recent reviews [182,183,184,185,186,187,188]. Actually, a recent systematic review, meta-analysis, and dose–response analysis of RCTs have shown that the chronic consumption of dark chocolate and flavanols increased the flow-mediated dilatation (FMD); also, the acute consumption of dark chocolate and both dark chocolate and flavanols had beneficial effects on the FMD. The consumption of more than 40 g/day of dark chocolate increases the FMD with the highest mean of FMD in doses around 40–60 g/day [176]. In addition, a more recent review and dose–response meta-analysis of RCTs indicated the beneficial effect of the acute and chronic consumption of cocoa-based products ingestion on platelet function and arterial stiffness in healthy adults, regardless of age and the pattern of consumption (4 weeks) in the chronic intake (4 weeks) and in the acute intake (120 min) [189]. However, not all data have been so positive. Last year, a meta-analysis restricted to diabetic patients suggested that there is weak evidence for a reduction in diastolic but not systolic blood pressure after mid/long-term cocoa flavanol administration. These effects seem stronger in females, younger, and hypertensive when cocoa flavanols are ingested in one daily intake and when the epicatechin content is high enough [190].

Within the very last years, some new data have been added to the general knowledge; thus, an acute crossover RCT was conducted in 20 healthy males, adding further evidence that epicatechin is a causal vasoactive molecule within flavanol-containing foods/beverages. Interestingly, the data revealed that intake levels as low as 0.5 mg/kg body weight are capable of inducing acute improvements in vascular function, evaluated as the FMD, in healthy volunteers [191]. In the same line, in a cohort of healthy young and elderly subjects, the twice-daily consumption of 450 mg of cocoa flavanols during a two week period decreased the endothelial compared to the cocoa flavanol-free control. This decrease is inversely correlated with FMD improvements, which indicates that flavanol consumption can improve the endothelial functional integrity in healthy humans [192]. Moreover, data from 25,618 participants of the European Prospective Investigation into Cancer (EPIC) Norfolk cohort showed that hypertensive participants had a stronger inverse association between flavanol biomarkers and systolic blood pressure when compared to normotensive participants. Therefore, flavanol intake could have a role in the maintenance of cardiovascular health in the general population [193]. In a trial published early this year, 20 healthy middle-aged men consumed cocoa flavanols (twice daily, 450 mg) or control drinks for 1 month, and the results showed that flavanol consumption may mediate the vascular protective effects by modulating the gene expression and DNA methylation and by preserving the integrity of the immunological–endothelial barrier functions [194]. Moreover, in this current year, an RCT with cocoa extract supplementation (500 mg flavanols, of which included 80 mg epicatechin, daily) among 21,442 US adults reported that cocoa extract supplementation did not significantly reduce the total cardiovascular events among older adults but reduced the cardiovascular disease death by 27% [195].

However, not all the studies have shown beneficial effects; thus, a recent meta-analysis showed a significant inverse association between cocoa consumption and systolic/diastolic blood pressure, but the analysis could not conclude any beneficial effect of cocoa consumption on blood pressure in normotensive/elevated blood pressure subjects [196]. Moreover, 200 mg daily of monomeric and oligomeric flavanols for three months on top of habitual diet and usual care did not reduce the plasma markers of endothelial dysfunction compared to the placebo in patients with long-term type 2 diabetes [197]. Finally, early this year, a crossover trial with type 2 diabetic (T2DM) and non-diabetic subjects that received 790 mg of cocoa flavanols daily reported that no beneficial effects of cocoa flavanols were detected on the vascular reactivity parameters in T2DM and non-diabetic participants [190].

## 4. Mechanisms of Action of Cocoa Methylxanthines

### 4.1. Recent Advances in Mechanisms of Action of Cocoa Theobromine

Methylxanthines show important biological activities that have several benefits for human health, acting against respiratory and cardiovascular diseases, cancer, obesity and diabetes, human infertility, neurological, and neurodegenerative diseases [198,199]. Theobromine (3,7-dimethylxanthine) is the main alkaloid of cocoa and its biological effects have been usually considered secondary to those of the main cocoa polyphenols, flavanols. However, recent studies in rats fed with cocoa or only theobromine have reported that this methylxanthine is the main factor responsible for cocoa’s effects on body weight gain as well as on lipid and glucose metabolism [200]. This and other results have raised interest in the study of the cocoa compound and synthetic chemical derivatives, such as pentoxifylline, formed by introducing a hexanone group to the theobromine molecule. This derivative is a non-selective phosphodiesterase inhibitor that has shown promising results, and it is being widely used as a therapeutic agent [201]. Some of the advances in the knowledge of the biological effects and mechanisms of action of theobromine are discussed below.

### 4.2. Recent Advances in Theobromine on Signaling Pathways

Many reported effects of theobromine are related to its role as a phosphodiesterase inhibitor. In high concentrations, which cannot be reached by nutritional intake but through pharmaceutical administration, theobromine and other methylxanthines lead to the inhibition of cyclic nucleotide phosphodiesterases and high-affinity ATP-dependent cyclic nucleotide transporters. This inhibition results in an increase in the cellular cyclic AMP levels and mobilization of intracellular calcium and, as a consequence, a modulation of the gamma-aminobutyric acid (GABA) receptors [202]. Still, in the last century, a pioneer study reported that theobromine acted as a purinergic receptor antagonist on tumor-induced angiogenesis in BALB/c mice [203]. The results of this study suggested that theobromine reduces neovascularization, tumor progression, and metastasis via the adenosine A2a receptor inhibition in ovarian cancer cells. Some years later, but still within the last century, another study described the influence of theobromine on the angiogenic activity and proangiogenic cytokines production of human ovarian cancer cells [204]. A later study showed that theobromine prevented malignant glioblastoma proliferation by negatively regulating phosphodiesterase-4, extracellular signal-regulated kinases (ERK), Akt/mTOR kinase, and nuclear factor-κB (NF-κB) [205]. A year later, new data suggested that theobromine inhibited the adipocyte differentiation during the early stages of adipogenesis by regulating the expression of peroxisome proliferator-activated receptor (PPAR)γ and CCAAT/enhancer-binding proteins (C/EBPs)α through the AMPK and ERK/JNK signaling pathways in the 3T3-L1 preadipocytes [206]. Simultaneously, theobromine was observed to increase the NAD+/SIRT1 activity protecting the kidney under diabetic conditions [207].

Two years later, a study reported that theobromine presented anti-proliferative activity against colorectal cancer cells HT-29 by modulating the gene expression associated with cell growth pathways. In fact, the Bax/Bcl-2 ratio was strongly upregulated in cells exposed to theobromine, significantly decreasing the cell proliferation [208]. The following year, Yoneda and colleagues showed that theobromine upregulates the cerebral brain-derived neurotrophic factor (BDNF) and augmented the cAMP/CREB/BDNF pathways and motor learning in mice [209]. Still, in 2017, theobromine suppressed the adipocyte differentiation and induced the C/EBPβ degradation by increasing its post-translational modification by sumoylation. Furthermore, it has been shown that the inhibition of AR1 signaling is important for theobromine-induced C/EBPβ degradation [210]. In a recent study, theobromine inhibited dimethylhydrazine-induced colon cancer through the downregulation of the Akt/mTOR and JAK2/STAT3 pathways and an increment of the Smad2 tumor suppressor [211]. Additionally, theobromine supplementation upregulated multiple thermogenic adipocyte marker genes in subcutaneous adipose tissue. Furthermore, in mouse-derived primary adipocytes, theobromine upregulated the expression of the uncoupling protein (UCP)1 and mitochondrial mass in a peroxisome proliferator-activated receptor γ (PPARγ) ligand-dependent manner, and it also increased the phosphorylation levels of the PPAR γ coactivator 1 α. These results indicate that dietary supplementation with theobromine induces browning in subcutaneous white adipose tissue and suggests its potential to treat obesity [212].

Theobromine downregulated the sterol regulatory element-binding protein (SREBP)-1c, fatty acid synthase (FASN), and upregulated the PPARα and CPT1a mRNA and protein levels in cultured hepatocytes and improved non-alcoholic fatty liver disease by inhibiting lipogenesis and fatty acid uptake and promoting fatty acid oxidation in the liver, which might be associated with its suppression of the mTOR signaling pathway [213]. Recent research provided evidence that methylxanthines can induce changes in lipid profiles that might be beneficial with respect to neurodegenerative diseases, such as Alzheimer’s and other diseases affected by lipid alterations [214]. Similarly, theobromine, from the cocoa shell, enhanced the ERK1/2 phosphorylation and fibroblast growth factor (FGF)21 release via PPAR activation. The increase in the phosphorylation of the insulin receptor, AKT, AMPK, mTOR, and ERK1/2 conduced to the regulation of hepatic mitochondrial function and energy metabolism [215]. Regarding the redox status and inflammatory process, a motivating study from 2015 in mice fed a theobromine-rich diet for 20 days showed that the main methylxanthine of cocoa is able to stimulate the Nrf2 activation and, consequently, the expression of both superoxide dismutase (SOD)-1 and GPx by itself [216]. More recently, theobromine displayed a preventive effect against interleukin (IL)-1β-induced chondrocyte dysfunction through the reduction in the IL-1β-induced production of cellular ROS and inflammatory mediators, including ciclooxigenase 2 (COX-2), prostaglandin E2, and NF-KB [217].

### 4.3. Recent Advances in Cocoa Theobromine on Cardiovascular Function

One of the first reports showed that the acute consumption of high-flavanol/high-theobromine chocolate increased the plasma epicatechin and theobromine concentrations and decreased the arterial stiffness, with no effect on the endothelial function and a marginal increase in the diastolic blood pressure. However, the chronic intake increased the plasma theobromine, though it did not have positive impacts on the endothelial function, arterial stiffness, or blood pressure in pregnant women at risk of preeclampsia [8]. On the other hand, the results show that the regular consumption of a theobromine/flavanols-rich cocoa product may contribute to the changes in cholesterol (and indirectly the HDL cholesterol) observed after the regular intake in healthy and cardiovascular risk subjects. The data also suggest that theobromine and 7-methylxanthine (the main theobromine metabolite), together with its content in insoluble dietary fiber, may be responsible for the decrease in IL-1β and the hypoglycemic effects observed [218]. In a study from last year, theobromine protected against the glutamate toxicity-induced GABAergic decline in the brain of rats with transient global cerebral ischemia injury. These findings suggested that theobromine could alleviate the chances of stroke by preventing acute episodes of cerebral ischemia [9].

Other authors observed a reduction in the brain oxidative stress, inflammatory intermediaries (tumor necrosis factor α (TNF-α), interleukin-1β, and -6, NF-κB), markers of cell damage (lactate dehydrogenase and caspase-3), acetylcholinesterase activity, and improvement in γ-aminobutyric acid quantity in rats that were given theobromine for 14 days daily after cerebral hypoperfusion [10]. Similarly, theobromine, as the most abundant component of a cocoa shell extract, showed vasodilator properties associated with increased NO bioavailability, suggesting that such a cocoa by-product is a potential fool ingredient for diseases related to endothelial dysfunction [219]. However, not all studies have supported the beneficial effects of theobromine or its derivatives. Thus, trying to decipher whether the beneficial effects of cocoa were partly due to theobromine, Talbot and coworkers [220] designed an RCT with 30 overweight and 14 obese men and women that consumed daily 500 mg of theobromine or placebo for 4 weeks. The goal was to unravel whether the effects of theobromine are mediated through the effects on HDL-mediated cholesterol efflux, which may be affected by the microRNA (miRNA) levels in the HDL particles. The results showed that theobromine did not improve the fasting and postprandial ATP-binding cassette subfamily A Member 1-mediated cholesterol efflux capacity but decreased the fasting miR-92a levels. The authors concluded that 4 weeks of theobromine consumption did not change the HDL-mediated cholesterol efflux capacity at baseline and postprandially [220]. In another study, theobromine consumption did not improve the fasting and postprandial endothelial function but increased the postprandial peripheral arterial diameters and decreased the augmentation index. These findings do not support the contribution of theobromine alone to the proposed cardioprotective effects of cocoa [221].

### 4.4. Recent Advances in Cocoa Theobromine on Cognitive Function and Other Effects

Until recently, cocoa’s effects on cognitive function were plainly associated with the polyphenolic fraction, especially flavanols [13]. In fact, most of the previous studies dealing with the effect of theobromine on cognitive and other brain functions failed to show significant changes [222]. However, it has been known for some years that theobromine blocks adenosine receptors to reduce the endogenous inhibitory adenosine and evoke a stimulatory effect on the CNS; thus, increased dopaminergic activity in the brain is suggested to mediate the psychostimulant effect [202]. This significant effect raised interest in the specific effects of theobromine independently from those already reported for flavanols.

The first encouraging results were published still in the last century. Pioneer studies showed that theobromine ingestion caused an increase in the brain levels of A1-adenosine receptors [223]. No more positive results were reported until ten years later when the administration of 30 mg/kg of theobromine to mice increased the ambulatory activity [224]. Some years later, in an animal experiment, Fernández-Fernández and coworkers [216] observed an enhanced modulatory effect on both cholinergic and catecholaminergic transmissions on mice fed 20 days with a theobromine-rich diet. The enhancing effect of theobromine on the levels of acetylcholine-related enzymes, dopamine, and especially noradrenalin confirms its beneficial role on the “cognitive reserve” and, consequently, a possible reducing effect on the cognitive decline underlying aging and Alzheimer’s disease [216]. In a study with volunteers to unravel the differential contributions of theobromine and caffeine on mood, psychomotor performance, and blood pressure, the authors concluded that caffeine might have more CNS-mediated effects on alertness, while theobromine might be acting primarily via peripheral physiological changes [225]. Moreover, in 2015, in a study of patients with Alzheimer’s disease, theobromine was found to be associated with a favorable Aβ profile in the cerebrospinal fluid [226].

Somewhat later, a study using a fat-enriched diet to induce a long-term deterioration in cognitive and memory functions showed that theobromine, at realistic concentrations in drinking water, restored the A1 receptor levels and improved the cognitive functions and Aβ levels [227]. Still, in 2017, the results in experimental animals strongly suggested that in mice orally administered 0.05% theobromine for 30 days, a phosphodiesterase inhibitor effect in the brain was produced, and it augmented motor learning through the cAMP/CREB/BDNF pathways (see also the section on signaling pathways), which play a crucial role in memory and learning [209]. In a comprehensive review from 2019 exploring the potential of theobromine as a cognitive regulator, the authors concluded that animal and human studies suggested a potential neuroprotective action of the long-term consumption of theobromine through a reduction in the Aβ amyloid pathology, which is observed in Alzheimer’s disease patients’ brains [222]. Early this year, treatment with theobromine significantly attenuated the neurological deficits and improved the sensorimotor functions and memory in rats with cerebral hypoperfusion. The authors suggested that the findings in the parameters regarding the oxidative stress, inflammatory intermediaries, and histopathological analysis substantiated the attenuation of neurodegenerative changes by theobromine [10]. Moreover, in this current year, a cross-sectional study on a representative American population aged ≥60 years has shown that a daily intake of theobromine tended to be associated with a better cognitive performance [11].

Regarding other biological effects recently attributed to theobromine, it is worth mentioning that methylxanthine plays a relevant role in some effects related to cocoa intake, such as the lower proportion of IgA-coated bacteria. Moreover, theobromine modifies the gut microbiota, although other cocoa compounds could also act on intestinal bacteria, attenuating or enhancing the theobromine effects [228]. Finally, research from this current year supports the use of methylxanthines, in particular theobromine, as a SARS-CoV-2 inhibitor as compared to chloroquine [229].

## 5. Conclusions

Cocoa bean processing is important as it improves both the economic and technological values of the beans. The primary and secondary processing of cocoa develops the characteristic flavor and color desired in chocolate and other cocoa-related products. However, processing, including chocolate production, affects both the composition and quantities of the flavanols and methylxanthines, mainly the theobromine. Cocoa bean processing mainly results in the reduction in the flavanols and theobromine content of cocoa beans and cocoa-related products. This modifies the taste and flavor as well as the health-promoting properties of cocoa beans and chocolates. Most of the work conducted on the influence of processing on the flavanols and methylxanthine contents in cocoa have focused mainly on the primary processes, with few processing steps during the industrial processing (mainly roasting and alkalization). No studies have reported on the impact of cocoa nib grinding, liquor pressing, and chocolate tempering on the composition of the bioactive compounds. Limited studies have been reported on the influence of conching. Considering the increased consumer interest in cocoa and cocoa-related products due to the presence of bioactive compounds, and the effect processing might have on their bioactivity and mechanisms of action, a holistic study of the entire processing steps (from beans to chocolates and other related products) and its impact on the bioactive compounds would have significant technological and commercial implications. Again, most of the studies reviewed here deal with cocoa beans, cocoa shells, cocoa powder, or black high-cocoa chocolates in which mainly flavanols and theobromine coexist. In this sense, it is challenging, in most cases, to attribute the activity and the mechanisms of action to one fraction or the other. However, we can conclude that theobromine seems to be effective, mainly through the regulation of the fatty acid metabolism, mitochondrial function, and energy metabolism pathways. Flavanols, however, seem to work through the protein kinases MAPKs and PI3K/Akt signaling pathways but also by regulating oxidative stress via targeting the transcription factor Nrf2 and Keap1, which participate in the activation of the antioxidant response element. Both flavanols and theobromine might be responsible for the nitric oxide regulatory effect of cocoa and dark chocolate for the neurological activity through this pathway and the regulation of neurotrophic factors and neuroplasticity pathways.

Taking all of the above into account, it would be interesting to investigate the effects that the transformation of cocoa into chocolate produces on the original cocoa bioactive compounds and on the molecular mechanisms associated with their intake. On the other hand, there is a clear need to search for a final product that eliminates, if not completely, most of the sugar content that chocolates currently have on the market. This would allow for enhancing the health benefits associated with the consumption of chocolate by eliminating the risk associated with the consumption of added sugars and by selectively increasing the bioactive compounds of interest by modifying the technological food processes.

## Figures and Tables

**Figure 1 ijms-23-14365-f001:**
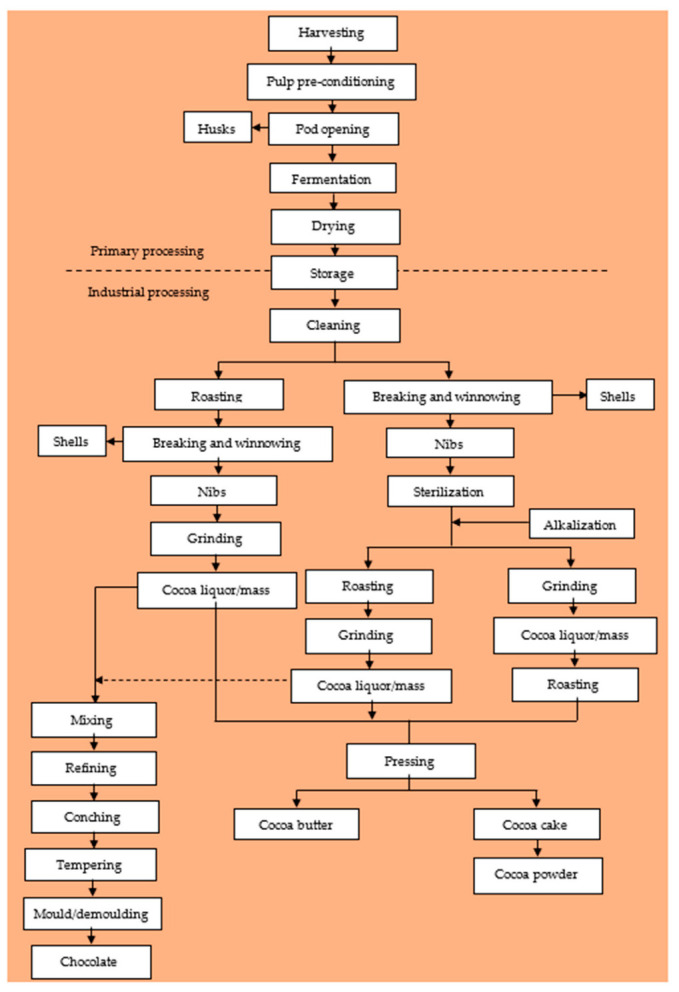
Schematic diagram of cocoa bean processing and chocolate production.

**Table 1 ijms-23-14365-t001:** Processing effects on polyphenolic and methylxanthines composition in cocoa beans and chocolate.

Stage	Process	Changes Occurring	Antioxidant Capacity	References
**Primary processing**	Harvesting	Increased proanthocyanidins, CF, and TBR with increasing pod maturation and ripening at harvest.	Increase in the antioxidant capacity with an increase in pod maturation and ripening.	[30,32]
	Pod storage	Reduction in TP, EC, C, TBR, and CF with increasing duration of pod storage.	Decrease in the antioxidant capacity with increasing duration of pod storage during fermentation.	[18,34,35,36]
	Fermentation	Decrease in the concentrations of EC, C, anthocyanins, phenolic acids, and TP with increasing duration of fermentation. Moreover, a decrease in the TBR and CF concentrations.	Decrease in the antioxidant capacity of cocoa beans with increasing duration of fermentation.	[18,22,23,37,38,39,40,41,42]
	Drying	Degradation of TP, EC, C, and anthocyanins. Decrease in the alkaloids (TBR and CF) content.	Reduction in the antioxidant capacity during the drying of cocoa beans.	[39,40,42,43,44,45]
**Industrial processing**	Roasting	Reduction in the amounts of TP, EC, procyanidin B2 and C1, anthocyanins, quercetin glycosides, TBR, and CF. An increase in C due to epimerization of EC.	A general reduction in the antioxidant capacity during cocoa bean roasting.	[22,23,30,40,41,42,45,46,47,48,49,50,51,52]
	Alkalization	Reduction in TP, EC, C, procyanidin B2 and C1, quercetin, quercetin-3-glucuronide, quercetin-3-arabinoside, isoquercetin, TBR, and CF.	Antioxidant activity is reduced.	[20,21,40,42,47,53,54]
	Conching	Decrease in TP during conching. No significant change in the polyphenol content during conching.	Decrease in the antioxidant capacity.	[52,55,56,57,58,59,60]

NB: TP—Total polyphenols; EC—Epicatechin; C—Catechin; TBR—Theobromine; CF—Caffeine.

## Data Availability

The data presented in this study are available in the article.

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
