# Peer review of "From Cocoa to Chocolate: Effect of Processing on Flavanols and Methylxanthines and Their Mechanisms of Action"

_ijms, 2022, doi:10.3390/ijms232214365_

Round 1

Reviewer 1 Report

Dear Authors,

Before publishing your paper, I encourage you to slightly correct and supplement it in accordance with the guidelines provided by me in the attached revised manuscript and the general comments that I include in this review:

1. I propose to extend the introduction section by pointing to specific transformations of phenolic compounds and methylxantines during processing and the resulting changes in their biological activity. It is worth using more than 8 references in this section in order to indicate the discrepancies in the results obtained by different authors. This chapter should also emphasize the purpose of the study by clearly indicating what the authors would like to demonstrate by reviewing the literature data on the given topic.

2. The conclusions in the current version are rather a summary of the considerations and it would be worth editing them so that they constitute a clear answer to the assumed, emphasized (as described above) purpose of the research. It is also worth including tips helpful in planning future research in this area (for example a scope that would particularly require analysis in the future).

3. Interpretation of some literature positions is too subjective, e.g. the paper states that in the reference number 145 the authors claim that "...cocoa flavanols should be considered the best well-established of all polyphenols", while they wrote that the influence of cocoa polyphenols on neuroprotective features requires further research. And that today the research on the biological activity of polyphenols derived, for example, from tea or olive oil, seems to be more promising.

4. The lack of electronic links to literature sources (in the References chapter) makes it difficult to read the content of the mentioned articles. It is also worth paying attention to other editorial inaccuracies, e.g. incorrect page numbering.

5. The paper should be checked in terms of English language. I am not an expert in this area, but I found the paper quite difficult to read due to the terms used by the authors. For example, in lines 560-562 they write about "... acute and chronic consumption", which should be defined rather as, for example, "a large amount of products consumed for a long time". The term "acute" is also used by the authors in relation to research (an acute crossover RCT ") or" acute improvements "and it is worth having a specialist to verify the correctness of these terms.

5. Frequent use of abbreviations by authors, e.g. the above-mentioned RCT, but also others, such as e.g. FMD, CVD or GABA, BDNF, UCP and FASN is also burdensome. It would be understandable if these terms would appear frequently in the text, but when abbreviations are numerous and they appear several lines or pages after their first use, finding explanations for the abbreviations is a great difficulty in reading the text.

Author Response

We want to thank you for your constructive and valuable observation, which undoubtedly help us to improve the quality of the paper. Below, we provide a point-by-point reply to the comments.

General comments

Dear Authors,

Before publishing your paper, I encourage you to slightly correct and supplement it in accordance with the guidelines provided by me in the attached revised manuscript and the general comments that I include in this review:

  1. I propose to extend the introduction section by pointing to specific transformations of phenolic compounds and methylxanthines during processing and the resulting changes in their biological activity. It is worth using more than 8 references in this section in order to indicate the discrepancies in the results obtained by different authors. This chapter should also emphasize the purpose of the study by clearly indicating what the authors would like to demonstrate by reviewing the literature data on the given topic.

Response: The introduction has now been expanded to address the issues raised. The introduction in the revised manuscript now reads as…

“Cocoa (Theobroma cacao L.) belongs to the family Malvaceae and the genus Theobroma. The cocoa tree produces fruits, referred to as pods along the trunk and branches. The pods are oval and contain the seeds (about 30–40 seeds), commonly known as cocoa beans, which are embedded in a sweet mucilaginous pulp. The pulp is rich in fermentable sugars and high in acidity (pH 3.0–3.5). Chocolate is the most popular cocoa-derived product highly valued by consumers around the world [1-3]. Its consumption is receiving increased global attention in recent years due to the biologically active components in cocoa beans [1-4]. However, some adverse effects of chocolate consumption have also been reported, as the final chocolate might be rich in sugar and fat [5]. This limits the use of chocolate as a functional food product and opens a search for more “natural” and un-processed forms of cocoa. The components of cocoa bean quality currently used on the international cocoa market include the bean size and mass, moisture content, flavour characteristics, low debris and bean defect levels, fat content, and quality [4,5]. There is a growing global trend in the search for functional attributes in food as quality indicators. In the case of cocoa beans, the final qualitative and quantitative content of polyphenols and methylxanthines might be considered as part of the quality indicators affecting the final price of the beans.

The present revision will focus on two different types of phytochemicals from cocoa, the main cocoa polyphenols, flavanols, and foremost cocoa methylxanthine, theobromine, since this compound is as abundant in cocoa as the total of flavonoids and is currently the subject of an increasing number of studies [8-11]. Cocoa flavanols and methylxanthines control molecular pathways that regulate cell signaling and function in most tissues so far investigated. The main pathways studied include the mitogen-activated protein kinase (MAPK) pathways and phosphoinositide-3-kinase-protein kinase B/protein kinase B (PI3K/Akt), which are activated by several ubiquitous growth factors such as insulin-like growth factor-1 (IGF-1) and fibroblast growth factor (FGF) among others, and neuro-specific factors such as brain-derived neurotrophic factor (BDNF) and nerve growth factor (NGF) [6,7]. Additionally, to the canonical signaling pathways, systemic factors and molecular/biochemical mechanisms such as nitric oxide (NO), toll-like receptors, apoptosis, antioxidant and anti-inflammatory responses are also specifically involved in the biological effects of cocoa phytochemicals on tissue and cell function and these will be discussed in this review. It is worth mentioning that most, if not all, of these pathways, are interconnected and it results very difficult most times to unravel and delineate the specific effects. Moreover, several polyphenols included in a diet may affect more than one biomarker or pathway, which may confound conclusions about the specificity of action [6-8].

Raw cocoa beans are bitter and astringent and need to be processed to reduce the bitterness and astringency and develop desired flavours. The bitterness and astringency of raw cocoa beans are due to the phenolics and methylxanthines in the beans [3]. During cocoa bean processing, a wide range of chemical reactions, including aldol condensation, polymerization, cyclization, Maillard reaction, and Strecker degradation, which enhance the flavour, colour, and shelf stability of cocoa products, have been reported [15-21]. According to some authors, the polyphenols undergo biochemical modification through polymerization and complexation with other compounds such as proteins [6, 16], while theobromine and caffeine mainly diffuse into the shells of the beans [22, 23]. These result in a decrease in the concentration of phenolic and methylxanthine compounds and contribute to the reduction in bitterness and astringency of the beans [22, 23]. However, there is no general agreement regarding the negative effect of food processing on the content of bioactive compounds in processed cocoa products. In fact, as occurs with other processed foods, some authors have shown that processing, far from reducing the biological activity of foods of plant origin, could enhance their bioavailability and, beyond that, their bioactivity [24,25]. Moreover, during processing, notably the fermentation and drying processes, flavour precursors (free amino acids and reducing sugars) are formed and these combine during the roasting process through a complex series of thermal reactions, mainly, the Maillard reaction and Strecker degradation and are converted into desirable aroma volatiles such as pyrazines, aldehydes, ketones, furans, pyrroles [18-21]. Nevertheless, cocoa processing clearly affects the flavanols and methylxanthine content of the final product and therefore will affect its potential biological activity and the molecular mechanisms that mediate it.

The purpose of this review is to provide a better understanding of the various processing steps of cocoa beans from harvesting to chocolate production emphasizing the changes in the main bioactive compounds in cocoa: polyphenols (mainly flavanols) and methylxanthines (mainly theobromine) during processing that will affect the mechanism of action implicated in their biological activity. Research gaps are also identified to aid in the planning of future studies. Since this review aims to be an update of the information regarding flavanols and methylxanthines of cocoa, most results reported up to 2020 will be briefly summarized and largely referred to comprehensive reviews; henceforth, we will mainly focus on recent data from the last couple of years, in which, despite the pandemic, research on this topic has been rather productive.”

  1. The conclusions in the current version are rather a summary of the considerations and it would be worth editing them so that they constitute a clear answer to the assumed, emphasized (as described above) purpose of the research. It is also worth including tips helpful in planning future research in this area (for example a scope that would particularly require analysis in the future).

Response: The conclusion has been revised to address the issues raised. The revised conclusion now read as it follows:

“Cocoa bean processing is important as it improves both the economic and technological values of the beans. Primary and secondary processing of cocoa develops the characteristic flavour and colour desired in chocolate and other cocoa-related products. However, processing, including chocolate production, affects both the composition and quantities of the flavanols and methylxanthines. Cocoa bean processing mainly results in the reduction of the flavanols and methylxanthine content of cocoa beans and cocoa-related products. This results in the modification of the taste and flavour as well as the health-promoting properties of cocoa beans and chocolates. Most of the works done on the influence of processing on the flavanols and methylxanthine contents in cocoa have focused mainly on the primary processes, with few processing steps during the industrial processing (mainly roasting and alkalization). No studies have reported on the impact of cocoa nib grinding, liquor pressing, and chocolate tempering on the composition of the bioactive compounds. Limited studies have been reported on the influence of conching. Considering the increased consumer interest in cocoa and cocoa-related products due to the presence of bioactive compounds and the effect processing might have on their bioactivity and mechanisms of action, a holistic study of the entire processing steps (from beans to chocolates and other related products) and its impact on the bioactive compounds would have significant technological and commercial implications. Again, most of the studies reviewed here deal with cocoa beans, cocoa shells, cocoa powder, or black-high cocoa chocolates in which mainly flavanols and theobromine coexist. In this sense, it is challenging, in most cases, to attribute the activity and the mechanism of actions to one fraction or the other. However, we can conclude that theobromine seems to be effective mainly through the regulation of fatty acid metabolism, mitochondrial function, and energy metabolism pathways. Whereas flavanols seem to work through the MAPKs and PI3K/Akt signaling pathways but also by regulating oxidative stress via targeting the transcription factor Nrf2 and Keap1, which participate in the activation of the ARE. Both flavanols and theobromine might be responsible for the NO regulatory effect of cocoa and dark chocolate for the neurological activity through this pathway and the regulation of neurotrophic factors and neuroplasticity pathways. Taking all of the above into account, it would be interesting to investigate the effects that the transformation of cocoa into chocolate produces on the original cocoa bioactive compounds and on the molecular mechanisms associated with their intake. On the other hand, there is a clear need to search for a final product that eliminates, if not completely, most of the sugar content that chocolates currently have on the market. This would allow for enhancing the health benefits associated with the consumption of chocolate by eliminating the risk associated with the consumption of added sugars and by selectively increasing the bioactive compounds of interest by modifying the technological food processes.”

  1. Interpretation of some literature positions is too subjective, e.g. the paper states that in the reference number 145 the authors claim that "...cocoa flavanols should be considered the best well-established of all polyphenols", while they wrote that the influence of cocoa polyphenols on neuroprotective features requires further research. And that today the research on the biological activity of polyphenols derived, for example, from tea or olive oil, seems to be more promising.

Response: We thank the reviewer for the relevant observation. We fully agree that the statement is exaggerated. We have therefore removed the quoted phrase from the new version of the manuscript.

  1. The lack of electronic links to literature sources (in the References chapter) makes it difficult to read the content of the mentioned articles. It is also worth paying attention to other editorial inaccuracies, e.g. incorrect page numbering.

Response: The list of references has been revised to include the electronic links (DOI) and the correct page numbers

  1. The paper should be checked in terms of English language. I am not an expert in this area, but I found the paper quite difficult to read due to the terms used by the authors. For example, in lines 560-562 they write about "... acute and chronic consumption", which should be defined rather as, for example, "a large amount of products consumed for a long time". The term "acute" is also used by the authors in relation to research (an acute crossover RCT ") or" acute improvements "and it is worth having a specialist to verify the correctness of these terms.

Response: We thank the reviewer for the comment. We have reviewed the new version of the manuscript in order to improve the English language. Although one of the authors is originally from an English-speaking country, we have also used another native English-speaking colleague to review the revised manuscript. On the other hand, we consider that most of the terms used can be understood by most of the International Journal for Molecular Science average readers.

  1. Frequent use of abbreviations by authors, e.g. the above-mentioned RCT, but also others, such as e.g. FMD, CVD or GABA, BDNF, UCP and FASN is also burdensome. It would be understandable if these terms would appear frequently in the text, but when abbreviations are numerous and they appear several lines or pages after their first use, finding explanations for the abbreviations is a great difficulty in reading the text.

Response: We have now double-checked the use of abbreviations to follow the Editorial rules

Specific comments

Line 42: Please provide examples of such studies

Response: Examples of such studies have been included in the revised manuscript [8-11]

Line 52: What does this term mean?

Response: The sentence has been revised to read as: “and these will be discussed in this review”.

Line 106: Sugars as a source of sugars?

Response: The sentence has been revised to read: “The harvested cocoa pods are living tissues and undergo metabolic activity such as respiration and transpiration using the sugars in the pulp.”

Line 112: Does it make sense to use an abbreviation for such a short and simple term?

Response: The abbreviation has been replaced with the full term: “pod storage”

Line 156: results?

Response: The sentence has been revised to read: “This results in the bleaching of the purple colour of the beans to brown….”

Line 187: Literature sources should be provided

Response: Literature sources have been provided. Kamphuis, 2009; Afoakwa, 2016

Line 198: Literature sources should be provided

Response: Literature source has been provided. Moser, 2015

Line 289: ?

Response: The sentence has been revised for clarity. It now reads as: “Cocoa liquor can be used directly in the manufacture of chocolate, or processed by pressing to obtain two main products – cocoa butter and cocoa pressed cake (with a fat content between 10–24%)”

Line 343: are?

Response: The sentence has been revised by replacing “is” with “are”

Line 368: What does this sentence mean? Have these authors viewed these works? And what did they show? Did they confirm their correctness or did they evaluate them?

Response: The sentence has been deleted

Line 396: plethora?

Response: We have added new references in order to justify the abundance of results regarding this last aspect.

Line 560-562: Can we speak of acute and chronic consumption? Isn't it better to talk about a large amount of products consumed for a long time?

Response: In fact, the meaning of acute does not imply a large amount of product. It means that the study was done after just one doses of it while chronic means that the ingestions of the compound or product of interest is more than ones

Line 591: CVD….What does it mean?

Response: The full meaning of CVD, cardiovascular disease has been included in the sentence now

Line 706: decipher….indicate?

Response: The word decipher has been replaced with the word “understand”. The sentence now reads… Thus, in trying to understand whether the beneficial effects of cocoa were….

Line 735: add the number of this reference

Response: The number of the reference, Fernández-Fernández and coworkers [] has been added

Line 747: Was there really a deliberate study using a high-fat diet to induce long-term cognitive and memory deterioration?

This study is in animals. In animals it is common to use high fat and sugar diets to induce neurological and cardiovascular disease models.

Line 763: use "this" or "current"

Response: The sentence has been revised to read… A current cross-sectional study on a representative American population aged ≥60 years has shown that daily intake of theobromine tended to be associated with better cognitive performance [220].

Line 786-790: I think it is worth not to use abbreviations in conclusions, so please replace them with the full names of specific compounds where possible.

Response: Thank you for the comment. The conclusion has been revised by removing all abbreviations

Reviewer 2 Report

1. article title should be revised and 

Kindly cite the following references

Di Mattia CD, Sacchetti G, Mastrocola D, Serafini M. From Cocoa to Chocolate: The Impact of Processing on In Vitro Antioxidant Activity and the Effects of Chocolate on Antioxidant Markers In Vivo. Front Immunol. 2017 Sep 29;8:1207. doi: 10.3389/fimmu.2017.01207. PMID: 29033932; PMCID: PMC5626833.

Å»yżelewicz, D., Budryn, G., Oracz, J., Antolak, H., KrÄ™giel, D., & Kaczmarska, M. (2018). The effect on bioactive components and characteristics of chocolate by functionalization with raw cocoa beans. Food research international (Ottawa, Ont.)113, 234–244. https://doi.org/10.1016/j.foodres.2018.07.017

Author Response

  1. article title should be revised

Response: Since Reviewer does not indicate in which sense he/she wants us to change the title we have decided to keep it us it was since we do not see how to improve it. Thank your for the comment since it has made us to double check it.

and 

Kindly cite the following references

Di Mattia CD, Sacchetti G, Mastrocola D, Serafini M. From Cocoa to Chocolate: The Impact of Processing on In Vitro Antioxidant Activity and the Effects of Chocolate on Antioxidant Markers In Vivo. Front Immunol. 2017 Sep 29;8:1207. doi: 10.3389/fimmu.2017.01207. PMID: 29033932; PMCID: PMC5626833.

Å»yżelewicz, D., Budryn, G., Oracz, J., Antolak, H., KrÄ™giel, D., & Kaczmarska, M. (2018). The effect on bioactive components and characteristics of chocolate by functionalization with raw cocoa beans. Food research international (Ottawa, Ont.)113, 234–244. https://doi.org/10.1016/j.foodres.2018.07.017

Response: The above references have been added accordingly

Reviewer 3 Report

This review aims to investigate the effect of processing on flavanols and methylxanthines and their mechanisms of action. This review clearly explained the process and steps of chocolate from cocoa, and it will provide a valuable contribution to the field when the paper is published. However, this paper needs minor revision before publishing.

Line #10 to #19: The authors need to state the major conclusion of flavanols and methylxanthines’ mechanisms of action in the abstract because it is one of the major subjects in this review paper.

Line #58: I would recommend explaining how bitter and astringent tastes are reduced and how desired flavours are developed from the processes in detail with the chemical aspect.

Line #83 and #98: The authors should strengthen the sentences by adding information on the mechanisms of how the flavanol and methylxanthine contents were increased.

Line #201: It would be great if the authors could describe what change in the flavanol and methylxanthine can occur after sterilization.

Line #277: I would suggest stating the possible change in the flavanol and methylxanthine after the grinding and liquor processing.

Line #301: Tempering is one of the most important processes for making proper chocolate. Are there any possible mechanisms that are related to flavanol and methylxanthine?
Line #334: What is MAPK? Is MAPK mitogen-activated protein kinase? Could you please write its full name first before using abbreviations? Please check other abbreviations as well.

Line #401, #622: I would recommend writing a short conclusion based on the authors’ review of other research at each section's end.

Line #800: The authors need to check the references again, ex) DOI for each reference.

Author Response

Reviewer 3

This review aims to investigate the effect of processing on flavanols and methylxanthines and their mechanisms of action. This review clearly explained the process and steps of chocolate from cocoa, and it will provide a valuable contribution to the field when the paper is published. However, this paper needs minor revision before publishing.

We thank you Reviewer 3 for the comments and suggestion. We have tried to address most of them accordingly.

Line #10 to #19: The authors need to state the major conclusion of flavanols and methylxanthines’ mechanisms of action in the abstract because it is one of the major subjects in this review paper.

Response: Thank you for the comment we have now included a general conclusion on the mechamism of action of cocoa bioactive compounds. We have also stated the limitations of the existing studies.

Line #58: I would recommend explaining how bitter, astringent tastes are reduced, and how desired flavours are developed from the processes in detail with the chemical aspect.

Response: We thank you for the comment. The sentence in line 58 has been revised to explain how astringent tastes and bitterness are reduced as well as how the desired flavours are developed. The sentence now reads:

“Raw cocoa beans are bitter and astringent and need to be processed to reduce the bitterness and astringency and develop desired flavours. The bitterness and astringency of raw cocoa beans are due to the phenolics and methylxanthines in the beans [3]. During cocoa bean processing, a wide range of chemical reactions, including aldol condensation, polymerization, cyclization, Maillard reaction, and Strecker degradation, which enhance the flavour, colour, and shelf stability of cocoa products, have been reported [15-21]. According to some authors, the polyphenols undergo biochemical modification through polymerization and complexation with other compounds such as proteins [6, 16], while theobromine and caffeine mainly diffuse into the shells of the beans [22, 23]. These result in a decrease in the concentration of phenolic and methylxanthine compounds and contribute to the reduction in bitterness and astringency of the beans [22, 23]. However, there is no general agreement regarding the negative effect of food processing on the content of bioactive compounds in processed cocoa products. In fact, as occurs with other processed foods, some authors have shown that processing, far from reducing the biological activity of foods of plant origin, could enhance their bioavailability and, beyond that, their bioactivity [24,25]. Moreover, during processing, notably the fermentation and drying processes, flavour precursors (free amino acids and reducing sugars) are formed and these combine during the roasting process through a complex series of thermal reactions, mainly, the Maillard reaction and Strecker degradation and are converted into desirable aroma volatiles such as pyrazines, aldehydes, ketones, furans, pyrroles [18-21]. Nevertheless, cocoa processing clearly affects the flavanols and methylxanthine content of the final product and therefore will affect its potential biological activity and the molecular mechanisms that mediate it..”

Line #83 and #98: The authors should strengthen the sentences by adding information on the mechanisms of how the flavanol and methylxanthine contents were increased.

Response: Additional information has been added to the sentences to strengthen the way in that flavanols and methylxanthine contents were increased. The sentence now reads as follows:

“A recent study by Dang and Nguyen [32] showed that maturity at harvesting and fermentation conditions significantly increased the flavanol and methylxanthine contents, together with the antioxidant capacity of the cocoa beans (Table 1). Earlier studies by Zheng et al. [33] and Pereira-Caro et al. [30] found theobromine and caffeine contents to appear at maturity when the cocoa beans started to develop their seed coats, which continued to increase until the cocoa fruits were fully ripe. Zheng et al. [33] also explained that the increase in alkaloid compounds could be due to their transport from the pericarp to the cocoa bean during fruit maturation, resulting in their increased concentration in the beans of matured fruits. Pereira-Caro et al. [30] noted that since alkaloid compounds were located in the same storage cell as phenolics, they could develop together with phenolics during maturation.”

Line #201: It would be great if the authors could describe what change in the flavanol and methylxanthine can occur after sterilization.

Response: For sterilization, an exposure time of seconds rather than minutes in the hot, low-pressure steam, is sufficient to sterilize the bean. This might be insufficient to cause any change in the flavanols and methylxanthine content. No work has reported changes in the flavanols and methylxanthine contents during sterilization. This statement has been added to the revised manuscript.

Line #277: I would suggest stating the possible change in the flavanol and methylxanthine after the grinding and liquor processing.

Response: Grinding cocoa nibs into liquor involves the application of shear under a controlled temperature. Changes in the polyphenols and methylxanthine contents during grinding have not been reported. Future studies have been proposed to focus on the effects of coarse and fine grinding of cocoa nibs on the polyphenols and methylxanthine composition of the final liquor produced. Again, the extent to which the bioactive compounds in cocoa liquor are affected during pressing remains unclear as no study has reported on it yet.

Line #301: Tempering is one of the most important processes for making proper chocolate. Are there any possible mechanisms that are related to flavanol and methylxanthine?
Response: Tempering is one of the most critical processing steps for making quality chocolate. It involves cooling under stirring the chocolate mass derived from conching (from 40–50 to 18–28 °C) to obtain a stable form of crystalline fat (polymorphic form V) responsible for the good melting properties and the glossy surface of good-quality chocolate [42,95]. However, changes in polyphenols and methylxanthine contents at this stage have not been reported. Studies are needed to understand the influence of tempering on the phenolic and methylxanthine composition of chocolates. A general statement has been included in the conclusion recommending the focus of future studies on these understudied processing steps.

Line #334: What is MAPK? Is MAPK mitogen-activated protein kinase? Could you please write its full name first before using abbreviations? Please check other abbreviations as well.

Response: We have added the full name the first time MAPK and all the other abbreviations are mentioned in the text following the recommendation that every word should be defined the first time its appear in each of three sections: the abstract; the main text; the first figure or table. When defined for the first time, the acronym/abbreviation/initialism has been added in parentheses after the written-out form.

Line #401, #622: I would recommend writing a short conclusion based on the authors’ review of other research at each section's end.

Response: we have added a conclusion that integrates all the results found by other authors as suggested. The final conclusion now reads:

Cocoa bean processing is important as it improves both the economic and technological values of the beans. Primary and secondary processing of cocoa develops the characteristic flavour and colour desired in chocolate and other cocoa-related products. However, processing, including chocolate production, affects both the composition and quantities of the flavanols and methylxanthines. Cocoa bean processing mainly results in the reduction of the flavanols and methylxanthine content of cocoa beans and cocoa-related products. This results in the modification of the taste and flavour as well as the health-promoting properties of cocoa beans and chocolates. Most of the works done on the influence of processing on the flavanols and methylxanthine contents in cocoa have focused mainly on the primary processes, with few processing steps during the industrial processing (mainly roasting and alkalization). No studies have reported on the impact of cocoa nib grinding, liquor pressing, and chocolate tempering on the composition of the bioactive compounds. Limited studies have been reported on the influence of conching. Considering the increased consumer interest in cocoa and cocoa-related products due to the presence of bioactive compounds, and the effect processing might have on their bioactivity and mechanisms of action, a holistic study of the entire processing steps (from beans to chocolates and other related products) and its impact on the bioactive compounds would have significant technological and commercial implications. Again, most of the studies reviewed here deal with cocoa beans, cocoa shells, cocoa powder, or black-high cocoa chocolates in which mainly flavanols and theobromine coexist. In this sense, it is challenging, in most cases, to attribute the activity and the mechanism of actions to one fraction or the other. However, we can conclude that theobromine seems to be effective mainly through the regulation of fatty acid metabolism, mitochondrial function, and energy metabolism pathways. Whereas flavanols seem to work through the protein kinases MAPKs and PI3K/Akt signaling pathways but also by regulating oxidative stress via targeting the transcription factor Nrf2 and Keap1, which participate in the activation of the antioxidant response element. Both flavanols and theobromine might be responsible for the nitric oxide regulatory effect of cocoa and dark chocolate for the neurological activity through this pathway and the regulation of neurotrophic factors and neuroplasticity pathways.

Taking all of the above into account, it would be interesting to investigate the effects that the transformation of cocoa into chocolate produces on the original cocoa bioactive compounds and on the molecular mechanisms associated with their intake. On the other hand, there is a clear need to search for a final product that eliminates, if not completely, most of the sugar content that chocolates currently have on the market. This would allow for enhancing the health benefits associated with the consumption of chocolate by eliminating the risk associated with the consumption of added sugars and by selectively increasing the bioactive compounds of interest by modifying the technological food processes.

.”

Line #800: The authors need to check the references again, ex) DOI for each reference.

Response: The list of references has been revised to include the electronic links (DOI) and the correct page numbers